


# Glacier thickness estimations of alpine glaciers using data and modeling constraints

Lisbeth Langhammer[1]
Melchior Grab[1,2]
Andreas Bauder[2]
Hansruedi Maurer[1]*

[1] Institute of Geophysics, ETH Zurich, Switzerland
[2] Laboratory of Hydraulics, Hydrology and Glaciology (VAW), ETH Zurich, Switzerland
* Corresponding author (*hansruedi.maurer@erdw.ethz.ch* )

## Abstract

Advanced knowledge of the ice thickness distribution within glaciers is of fundamental
importance for several purposes, such as water resource management and studying the
impact of climate change. Ice thicknesses can be modeled using ice surface features, but
the resulting models can be prone to considerable uncertainties. Alternatively, it is
possible to measure ice thicknesses, for example, with ground-penetrating-radar (GPR).
Such measurements are typically restricted to a few profiles, with which it is not possible
to obtain spatially unaliased subsurface images. We developed the Glacier Thickness
Estimation algorithm (GlaTE), which optimally combines modeling results and measured
ice thicknesses in an inversion procedure to obtain overall thickness distributions.
Properties and benefits of GlaTE are demonstrated with three case studies performed on
different types of alpine glaciers. In all three cases, subsurface models could be found
that are consistent with glaciological modeling and GPR data constraints. Since acquiring
GPR data on glaciers can be an expensive endeavor, we additionally employed elements
of sequential optimized experimental design (SOED) for determining cost-optimized
GPR survey layouts. The calculated benefit-cost curves indicate that a relatively large
amount of data can be acquired, before redundant information is collected with any
additional profiles and it becomes increasingly expensive to obtain further information.
Only at one out of the three test sites this level was reached.



## 1 Introduction

Estimating the amount of the glacier ice around the globe is crucial, for example, for sea-level predictions, securing fresh water recourses, designing hydropower facilities in high-alpine environments, and predicting the occurrence of glacier-related natural hazards. For estimating the overall glacier ice mass and its local distribution, (i) knowledge of the glacier outline, (ii) its surface topography and (iii) the underlying bedrock topography is required. The first two quantities can be observed with aerial and satellite imagery, but the bedrock topography is more difficult to determine.

The conceptually simplest option includes drilling boreholes through the glacier ice (e.g., Iken, 1988). This approach offers ground-truth information, but only a very sparse observation grid can be obtained with realistic efforts. Therefore, geophysical methods have been employed for obtaining more detailed information. Due to the very high electrical resistivity of glacier ice and the relatively high electromagnetic impedance contrast between ice and bedrock material, ground-penetrating-radar (GPR) techniques, also referred to as radio-echo-sounding (RES), have been the primary choice for such investigations (e.g., Evans, 1963). GPR data can either be acquired ground-based (e.g., Watts and England, 1976), or, more efficiently, using fixed-wing airplanes (e.g., Steinhage et al., 1999) or helicopters (e.g., Rutishauser et al., 2016).

Despite the powerful capabilities of modern GPR acquisition systems, it is still beyond any practical limits to acquire spatially un-aliased 3D data sets. GPR data are therefore collected only along a sparse network of profiles, which leaves considerable uncertainties in the regions between the profiles.

To address this problem, glaciological modeling techniques have been established to relate observable surface parameters to the thickness distribution of ice. One of the earliest concepts was published by Nye (1952). He established a simple relationship between the surface slope and ice thickness. During the past decades, more sophisticated ice thickness modeling techniques have emerged rapidly. Various glaciological constraints, such as mass conservation and/or the relation between basal shear stress and ice thickness, were considered (e.g., Farinotti et al., 2009;Huss and Farinotti, 2012;Clarke et al., 2013;Linsbauer et al., 2012;Morlighem et al., 2011). See Farinotti et al. (2017) for a more complete review of most of the approaches published to date.

Due to inaccuracies of the observed data (GPR measurements, surface topography, etc.) and/or inadequacies of the modeling approaches, modeled ice thicknesses cannot be expected to be perfect. This can be considered by formulating ice thickness estimation as an optimization problem, in which the discrepancies between observed and predicted data are minimized (e.g., Morlighem et al., 2014). In this contribution, we follow an approach similar to Morlighem et al. (2014), but with a different implementation. We introduce the general framework of Glacier Thickness Estimation (GlaTE), with which modeling and data constraints can be combined in an appropriate fashion. After introducing the underlying theory, we demonstrate the performance of the GlaTE inversion procedure with three case studies. In the second part of the paper, we employ elements of GlaTE to

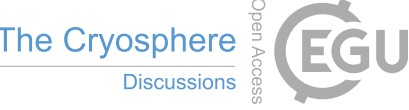



address the experimental design problem. Here, we seek a measured data set that offers
maximum information content at minimal costs. For that purpose, we consider
sequentially optimized experimental design (SOED) techniques (e.g., Maurer et al.,
2017). The paper concludes with a critical review of potential problems and shortcomings
of GlaTE and the associated SOED procedures, and we outline options to address these
issues and propose useful extensions of the methodology.
## 2 GlaTE inversion algorithm
### 2.1. Theory
The basic idea of GlaTE inversions is to combine observable data with glaciological
modeling constraints, whereby it is attempted to consider appropriately the uncertainties
associated with both types of information. All constraints are formulated, such that they
can be integrated into a single system of equations, which can be solved with an
appropriate solver.
The first type of constraints includes the GPR data. They can be written in the form of
(1)    $\mathbf{G}\mathbf{h}^{\mathbf{est}} = \mathbf{h}^{\mathbf{GPR}}$ ,
where $\mathbf{h}^{\mathbf{est}}$ is a vector including the unknown (*est*imated) ice thicknesses at M locations
(typically defined on a regular grid $R$ on a glacier), and $\mathbf{G}$ is a $N^{GPR} \times M$ matrix with
ones in its main diagonal and zeros everywhere else ($N^{GPR}$ = number of available GPR
data points, $M$ = number of elements in $\mathbf{h}^{\mathbf{est}}$). The vector $\mathbf{h}^{\mathbf{GPR}}$ of length $N^{GPR}$ includes the
GPR-based thickness estimates. Since the GPR data usually do not coincide with the grid
points of $R$, the values $\mathbf{h}^{\mathbf{GPR}}$ are obtained by interpolating or extrapolating the GPR data
to the nearest grid points of $R$.
Next, we consider glaciological modeling constraints. In principle, any of the algorithms
proposed in the literature can be employed. Here, we follow closely the approach
described in Clarke et al. (2013). Input data include a digital terrain model (DTM,
defined on $R$) and the glacier outline.
First, the glacier area is subdivided into so-called flowsheds using the Matlab TOPO-
Toolbox (Schwanghart and Kuhn, 2010). The subsequent procedure is applied to each
flowshed individually (see comments in Clarke et al. (2013) for more information on the
flowshed subdivision).
Next, the apparent mass-balance, defined as
(2)                                        $\tilde{\mathbf{b}} = \dot{\mathbf{b}} - \dfrac{\partial \mathbf{h}}{\partial t}$ ,



with $\dot{\mathbf{b}}$ being the mass balance rate, and $\dfrac{\partial \mathbf{h}}{\partial t}$ the thickness change rate, is either
determined by measuring $\dot{\mathbf{b}}$ and $\dfrac{\partial \mathbf{h}}{\partial t}$, or computed via the condition

(3)                                  $$\int_{\Omega_G} \tilde{\mathbf{b}} = 0 \; ,$$

where $\Omega_G$ denotes the glacier area (see Farinotti et al. (2009) for more details). In a next
step, the flowsheds are partitioned into a prescribed number of elevation zones $D_i$
($i = 1…number\ of\ elevation\ zones$), for which the ice discharge $Q_i$ through its lower
boundary is computed using

(4)                                  $$Q_i = \int_{\Omega_{D_i}} \tilde{\mathbf{b}} \; ,$$

where $\Omega_{D_i}$ is the area of zone $D_i$. Following Clarke et al. (2013), the basal shear stress $\tau$
can then be obtained via the relationship

(5)                          $$\boldsymbol{\tau} = \left[ \frac{(n+2)\rho g \sin(\phi)^2 \xi \mathbf{q}}{2A} \right]^{1/(n+2)}$$

The parameters $n$, $\rho$, g and $A$ denote the exponent of Glen's flow law, ice density, gravity
acceleration and creep rate factor, respectively (e.g., Cuffey and Patterson, 2010). The
factor $\xi$ denotes the creeping contribution (relative to basal sliding) to the ice flux
$(0 < \xi < 1)$, and $\mathbf{q}$ is the specific ice discharge $q_i = \bar{Q}_i / l_i$, where $l_i$ is the length of the
lower boundary of $D_i$, and $\bar{Q}_i$ is the average of $Q_i$ within $D_i$. Likewise, the angle $\phi$
represents the surface slope averaged along the lower boundary of $D_i$.

As outlined in Kamb and Echelmeyer (1986), the physics of ice flow can be incorporated
into the modeling procedure by applying "longitudinal averaging" of the shear stress (i.e.,
along the flow direction). We apply this procedure to the results obtained with
Equation (5). Finally, the ice thicknesses $\hat{\mathbf{h}}^{\mathbf{glac}}$ ($glac$ stands for glaciological modeling
constraints) are obtained using

(6)       $$\hat{\mathbf{h}}^{\mathbf{glac}} = \frac{\tau^*}{\rho g \sin(\theta)} \; ,$$

where $\boldsymbol{\tau}^*$ denotes the basal shear stress after longitudinal averaging.



Some of the parameters in Equation (5) may be subject to considerable uncertainties. For
example, the parameter $\xi$ is often poorly known, and it is not guaranteed that the values
of the parameters $A$ and $n$, usually taken from the literature, are accurate. Typically, $n$ is
reasonable well constrained, but A can vary over orders of magnitudes. Therefore, the
overall magnitudes of $\hat{\mathbf{h}}^{\mathbf{glac}}$ may be significantly over- or under-estimated. This can be
considered with an additional factor $\alpha_{GPR}$, yielding

(7) $\qquad \mathbf{h}^{\mathbf{glac}} = \alpha_{GPR}\hat{\mathbf{h}}^{\mathbf{glac}}$ .

$\alpha_{GPR}$ can be computed with an optimization procedure that minimizes
$\left| mean\left( \mathbf{h}^{\mathbf{GPR}} - \alpha_{GPR}\hat{\mathbf{h}}^{\mathbf{glac}} \right) \right|$ .

The correction factor $\alpha_{GPR}$ accounts for some inadequacies of Equation (5), but it is still
possible that there are systematic differences between $\mathbf{h}^{\mathbf{GPR}}$ and $\mathbf{h}^{\mathbf{glac}}$. To avoid the
resulting inconsistencies, we consider not the absolute values $\mathbf{h}^{\mathbf{glac}}$, but the spatial
gradients $\nabla\mathbf{h}^{\mathbf{glac}}$ as glaciological constraints, resulting in

(8) $\qquad \mathbf{L}\mathbf{h}^{\mathbf{est}} = \nabla\mathbf{h}^{\mathbf{glac}}$ ,

where $\mathbf{L}$ is a difference operator of dimension $M \times M$ .

Further constraints can be imposed via the glacier boundaries that can be determined
from aerial or satellite images or ground observations. They are considered in the form of
the equation

(9) $\qquad \mathbf{B}\mathbf{h}^{\mathbf{est}} = 0$ ,

where $\mathbf{B}$ is a $M \times M$ matrix with ones at appropriate places in its main diagonal.

Depending on the discretization of the glacier models (i.e., the discretization of $R$), the
constraints described above, may allow the resulting system of equations to be solved
unambiguously. However, in most cases, there will be still a significant underdetermined
component, that is, there will be many solutions that explain the data equally well. This
requires regularization constraints to be applied (e.g., Menke, 2012). A common strategy
for regularizing such problems is to follow the Occam's principle, which identifies the
"simplest" solution out of the many possible solutions (Constable et al., 1987). Here, we
define "simplicity" in terms of structural complexity, that is, we seek a smooth model.
This can be achieved via a set of smoothing equations of the form

(10) $\qquad \mathbf{S}\mathbf{h}^{\mathbf{est}} = 0$ ,

where $\mathbf{S}$ is a $M \times M$ smoothing matrix.




All the constraints can now be merged into a single system of equations

205    (11)
$$
\begin{pmatrix} \lambda_1 \mathbf{G} \\ \lambda_2 \mathbf{L} \\ \lambda_3 \mathbf{B} \\ \lambda_4 \mathbf{S} \end{pmatrix} \mathbf{h}^{\mathbf{est}} = \begin{pmatrix} \lambda_1 \mathbf{h}^{\mathbf{GPR}} \\ \lambda_2 \nabla \mathbf{h}^{\mathbf{glac}} \\ 0 \\ 0 \end{pmatrix},
$$


where the parameters $\lambda_1$ to $\lambda_4$ allow a weighting according to the confidence into
individual contributions. Parameter $\lambda_3$ is not critical and can be fixed to an appropriate
value (e.g., 1.0). The magnitudes of the remaining three parameters must be chosen, such
that the system of equations in (11) is solvable. However, it also needs to be considered
that all the constraints related to $\lambda_1$, $\lambda_2$ and $\lambda_4$ may be subject to significant inaccuracies.
It is difficult to predict the accuracy of the modeling constraints and to judge the
appropriateness of the smoothing constraints, but the accuracy of the GPR data
constraints, subsequently denoted as $\boldsymbol{\varepsilon}^{\mathbf{GPR}}$, can usually be quantified. Therefore, $\lambda_1$, $\lambda_2$
and $\lambda_4$ have to be chosen, such that the discrepancy of the GPR data $\left( \left\| \mathbf{G}\mathbf{h}^{\mathbf{est}} - \mathbf{h}^{\mathbf{GPR}} \right\| \right)$ is
of the order of $\boldsymbol{\varepsilon}^{\mathbf{GPR}}$, and the GPR data are thus neither under- nor over-fitted. We have
implemented this by choosing the magnitudes of $\lambda_1$, $\lambda_2$ and $\lambda_4$, such that a prescribed
percentage of the GPR data (e.g., 95%) satisfies $\left\| \mathbf{G}\mathbf{h}^{\mathbf{est}} - \mathbf{h}^{\mathbf{GPR}} \right\| < \boldsymbol{\varepsilon}^{\mathbf{GPR}}$.

This can be achieved with different strategies. One option is to fix $\lambda_2$ and $\lambda_4$, and to
vary $\lambda_1$ until the condition, mentioned above, is met. Alternatively, it is possible to fix
the pairs $\lambda_1 / \lambda_4$ or $\lambda_1 / \lambda_2$ and to vary $\lambda_2$ or $\lambda_4$. Choice of the most appropriate strategy
depends on the uncertainties associated with the individual contributions in Equation (11)

224    .


The dimension of the system of equations in (11) can be very large, but the matrices $\mathbf{G}$,
$\mathbf{L}$, $\mathbf{B}$ and $\mathbf{S}$ are all extremely sparse. Therefore, sparse matrix solvers, such as LSQR
(Paige and Saunders, 1982) can solve such systems efficiently for $\mathbf{h}^{\mathbf{est}}$.


**2.2 Performance tests**

For testing the GlaTE inversion algorithm, we investigated glacier ice thickness at three
sites in the Swiss Alps (Figure 1). The first site is Morteratschgletscher (Figure 1a).
Lying at altitudes between 2050 and 4000 m a.s.l. (Zekollari et al., 2013), the glacier has
a typical valley-glacier shape and is located in the Engadin region of Switzerland. In
2015, the tributary glacier Vadret Pers in the east detached from the main trunk of
Morteratschgletscher, but we continue to treat both glaciers as a connected system, since



the last available outline of the glaciers in 2015 shows the remnant of the former
connection. In 2010, the glacier system covered an area of $\approx$ 15 km$^2$, and it had a length
of $\approx$ 7.4 km.
The second site, Glacier Plaine Morte (2400-3000 m a.s.l., (Figure 1b), is the largest
plateau glacier in the European Alps (Huss et al., 2013). The surface slope is shallow
with slope angles less than 4° and a short glacier tongue draining towards the North.
The third site is a cluster of small valley flank and cirque-type glaciers on the eastern
flank of the Matter valley (Figure 1c) below the Dom peak. From North to South, the
glaciers are named Hohbärggletscher, Festigletscher, Kingletscher and
Weingartengletscher. The Hohbärggletscher is the largest (2800-4500 m a.s.l.) and
longest of the group.
For all sites, the recorded GPR profiles are shown in Figure 1. The GPR data are a
composite of several campaigns. Most of the data were recorded with the dual
polarization system AIR-ETH (Langhammer et al., 2018). On the Glacier Plaine Morte, a
grid of profiles was acquired in 2016, and on the Morteratschgletscher and in the Dom
Region in 2017. The data were processed as described in Grab et al. (2018), and the
bedrock depths and the corresponding ice thicknesses were obtained from the migrated
GPR images.
As input data for the glacier models, surface topography and an outline of the individual
glaciers was required. As surface topography, we used the swissALTID3D (DTM, Digital
Terrain Model Release 2017 © swisstopo (JD100042)). The most recent version covering
the individual glaciers was extracted and down-sampled to 10 m resolution. The outline
represents the extension of the glacier in 2015-2016. DTM and glacier outlines are
displayed in Figure 1.





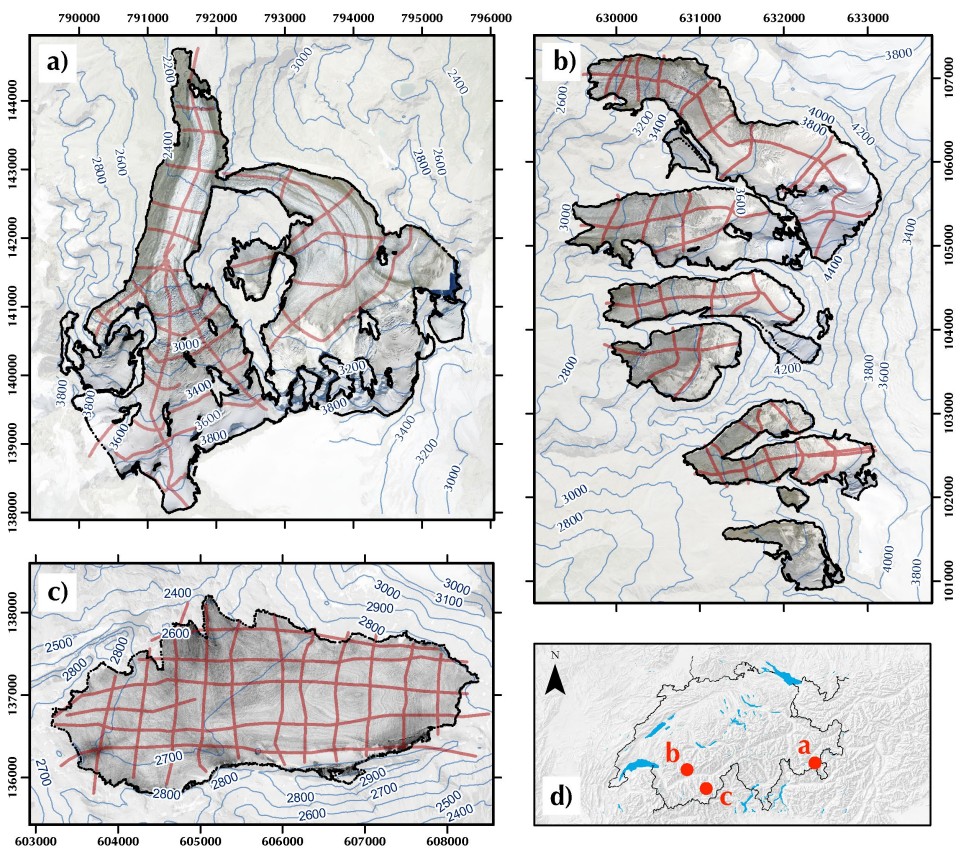

Figure 1: Satellite images and surface topography isolines of the glaciers investigated.
(a) Morteratschgletscher, (b) Glacier Plaine Morte and (c) Dom region. The Swiss map
in the bottom right panel indicates the locations of the glaciers. GPR profiles acquires
are shown in red. Orthophotos © 2017 swisstopo (JD100042). Coordinate system:
CH1903.

Before applying GlaTE inversions to all field sites, we tested the different options for
determining $\lambda_1$, $\lambda_2$ and $\lambda_4$, using the data from Morteratschgletscher. Figure 2 shows
the ice thicknesses distributions, (i) when only glaciological constrains are applied ($\mathbf{h}^{\mathbf{glac}}$,
Figure 2a), and (ii) when only GPR constraints are considered ($\mathbf{h}^{\mathbf{GPR}}$, Figure 2b). In the
latter case, the thicknesses are obtained by natural neighbor interpolation from the GPR
data. Since no extrapolation was performed, not all glacierized regions have an ice
thickness estimate. Both images exhibit increased thicknesses in the western glacier, but
only the glaciological constraints indicate an overdeepening in the eastern one, thereby
indicating that the two models are inconsistent.





Figure 3 shows the results of the GlaTE inversions using either prescribed $\lambda_1 / \lambda_4$ (Figure
3a), $\lambda_1 / \lambda_2$ (Figure 3c) or $\lambda_2 / \lambda_4$ (Figure 3e) pairs. The corresponding difference plots
(Figure 3b, d and f) refer to the deviation of the obtained thickness results compared with
the thickness calculated with the glaciological approach. We varied the $\lambda_2$ and $\lambda_4$
parameters by starting with very high values of 50, and by decreasing them successively
until 95% of the GPR data met the condition $\left\| \mathbf{Gh}^{\mathrm{est}} - \mathbf{h}^{\mathrm{GPR}} \right\| < \mathbf{\epsilon}^{\mathrm{GPR}}$, where $\mathbf{\epsilon}^{\mathrm{GPR}}$ was
estimated to be 5 m. In contrast, we started with a low value of 0.02 for variable $\lambda_1$, and
increased it successively until 95% of the data were fitted within the error $\mathbf{\epsilon}^{\mathrm{GPR}}$. Table 1
summarizes the prescribed and estimated $\lambda$ values.
All three inversion strategies (i.e., either varying $\lambda_2$, $\lambda_4$ or $\lambda_1$) yielded comparable
results. Although the difference plots with respect to the glaciological model exhibit
considerable differences (Figures 3b, 3d and 3f), the general shapes obtained with the
glaciological constraints were well preserved in regions where the GPR data coverage
was poor. From this first test, we conclude that (i) the GlaTE inversion approach works
well, and (ii) that the strategy by which the values of $\lambda$ are chosen is not critical.

| Inversion type | $\lambda_1$ | $\lambda_2$ | $\lambda_3$ | $\lambda_4$ | Figures |
|---|---|---|---|---|---|
| $\lambda_1 / \lambda_4$ fixed | 1 | 0.78 | 1 | 10 | 3a, 3b 4c, 4d |
| $\lambda_1 / \lambda_2$ fixed | 1 | 1 | 1 | 7.8 | 3c, 3d |
| $\lambda_2 / \lambda_4$ fixed | 1.28 | 1 | 1 | 10 | 3e, 3f |
| $\lambda_1 / \lambda_4$ fixed | 1 | 1.56 | 1 | 2 | 4a, 4b |
| $\lambda_1 / \lambda_4$ fixed | 1 | 0.00 | 1 | 50 | 4e, 4f |

*Table 1: Weighting parameters $\lambda$ employed for the GlaTE inversions shown in Figures 3*
*and 4. Numbers marked red indicate varying parameters.*
It is instructive to study the effects of an overly small or large (fixed) $\lambda_4$ value. As shown
in Table 1, we employed a prescribed value of 10 for $\lambda_4$. This value was chosen by trial
and error. There was a range of $\lambda_4$ values around 10 that yielded similar results (not
shown). Choosing very low or high $\lambda_4$ values (i.e., $\lambda_4 = 2$ resp. $\lambda_4 = 50$) has a
detrimental effect on the results, as shown in Figure 4. For $\lambda_4 = 2$, the inversion fits the
ice thicknesses obtained from the GPR data only along the profile lines and maintains the
glaciological modeling results in the remaining areas. This produces artificial features in
the thickness map (Figure 4a). In contrast, $\lambda_4 = 50$ produces overly smooth images,
which is obscuring small-scale variations from the glaciological constraints in regions

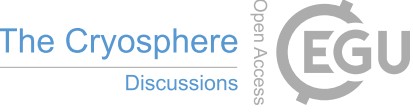

poorly covered by GPR data (Figure 4e). It is also noteworthy that even with $\lambda_2 = 0$ only
approx. 70% of the discrepancies $\left\| \mathbf{Gh}^{est} - \mathbf{h}^{GPR} \right\|$ were below $\boldsymbol{\varepsilon}^{GPR}$ (Figure 5e).


*Figure 2: Results from Morteratschgletscher only using (a) glaciological constraints and*
*(b) GPR constraints. Colors indicate ice thickness. Available thickness data obtained*
*from GPR profiles are marked with black lines.*





*Figure 3: Results from Morteratschgletscher using different strategies for choosing*
*weighting parameters λ (see text for more explanations). Left panels show ice thickness*
*distributions and right panels show differences to glaciological model without GPR*
*constraints (Figure 2a).*

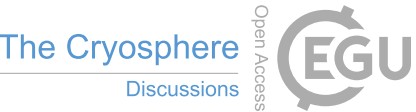

*Figure 4: Results from Morteratschgletscher using fixed $\lambda_1$ and $\lambda_4$ values and varying*
*$\lambda_2$. (a) and (b) are the results for $\lambda_4 = 2$, (c) and (d) for $\lambda_4 = 10$ and (e) and (f) for*
*$\lambda_4 = 50$. Left panels show ice thickness distributions and right panels show differences to*
*glaciological model without GPR constraints (Figure 2a).*




In the following, we consider only the scheme in which $\lambda_1 / \lambda_2$ is kept fixed
$(\lambda_1 = 1, \lambda_2 = 1)$, and $\lambda_4$ is varied for analyzing the Glacier Plaine Morte and the Dom
region data. The results are shown in Figures 5 and 6. For the Glacier Plaine Morte, the
glaciological model suggests a deep isolated trough slightly east of the center (Figure 5a).
This is not supported by the GPR data, which rather indicate a larger E-W oriented
elongated zone of increased thickness (Figure 5b). Such a feature is also contained in the
GlaTE inversion results (Figure 5c). Furthermore, the glaciological model in Figure 5a
overestimates the ice thickness in the northeastern part of the glacier.

Results from the Dom region show a relatively good match between the glaciological
model (Figure 6a) and the GlaTE inversion result (Figure 6c). The glaciological model
tends to underestimate the maximum thickness in the center of the glacier tongues, and to
overestimate the thickness towards the edges (Figure 6d). The isolated trough structures
(ice thickness > 200 m) in the northernmost glacier in the glaciological model (Figure 6a)
are only partially supported by the GPR data (Figure 6b) and the GlaTE inversion (Figure
6c). In the southernmost Weingartengletscher, no data constraints exist (Figure 6b). The
non-zero differences in this part (Figure 6d) are the result of the smoothing constraints.
Here, the thickness estimates from the glaciological model are thus more trustworthy.

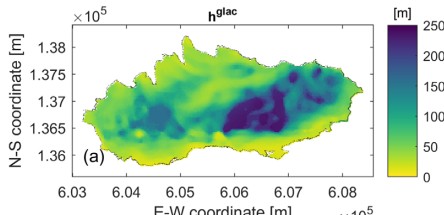
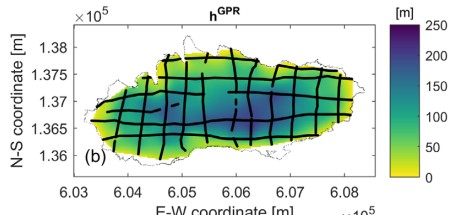
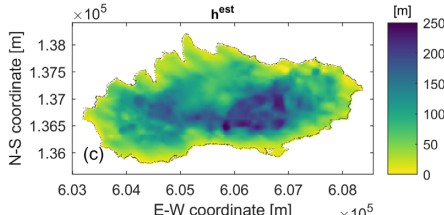
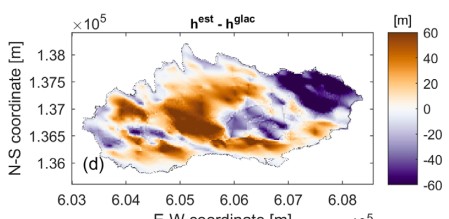

*Figure 5: Results from Plaine Morte Glacier. (a) only glaciological constraints, (b) only*
*GPR constraints (available thickness data from GPR profiles marked with black lines),*
*(c) GlaTE inversion, (d) difference between GlaTE inversion and glaciological model.*





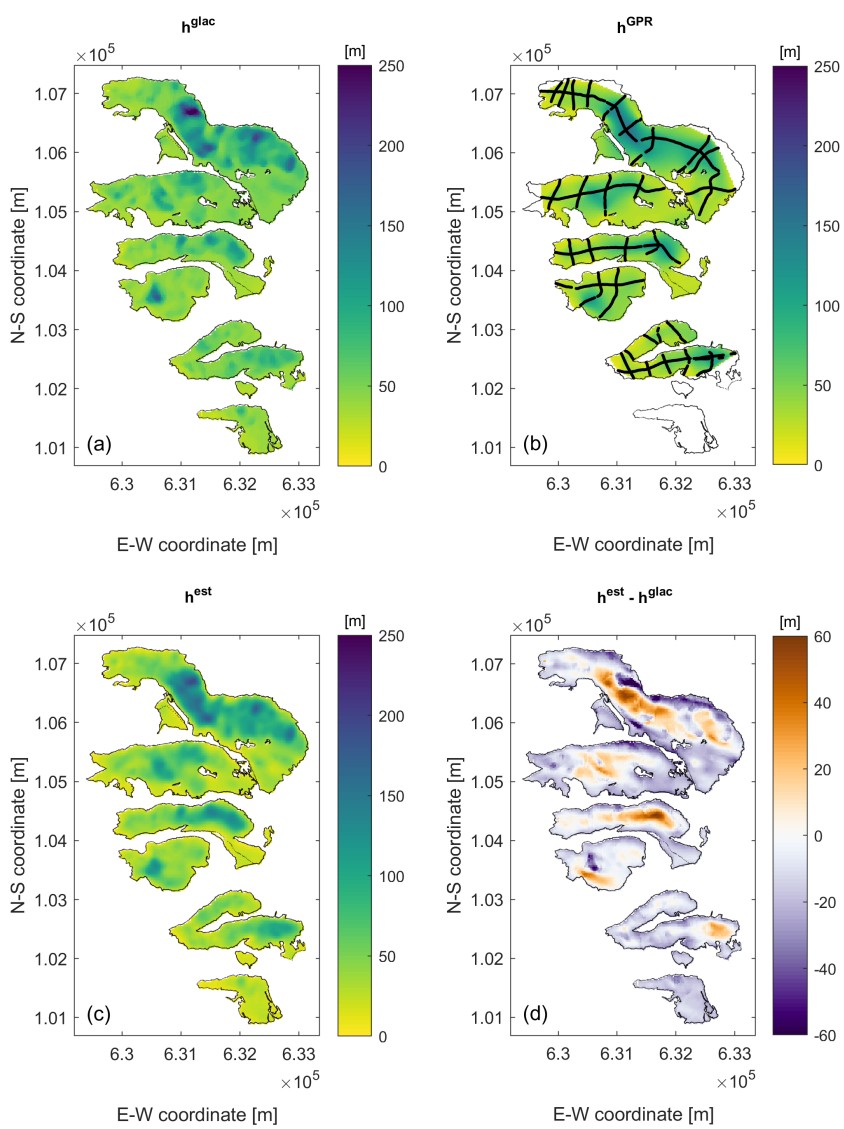

*Figure 6: Results from the Dom region. (a) only glaciological constraints, (b) only GPR*
*constraints (available thickness data from GPR profiles marked with black lines), (c)*
*GlaTE inversion, (d) difference between GlaTE inversion and glaciological model.*




## 3 Optimized experimental design using GlaTE inversion

All the investigations, described in Section 2, were based on existing GPR data. Their experimental layouts were designed heuristically using experience from prior surveys. Once a glacier model has been established, one may realize that another GPR survey layout may have provided better information. Therefore, a dense survey grid, as employed for 3D seismic reflection campaigns for hydrocarbon exploration for example (e.g., Vermeer, 2003) would be the best choice. This, however, would exceed by far the budgets typically available for glacier investigations.

Optimizing the glaciological constraints with only a limited number of GPR data is a chicken-and-egg problem: identifying the most useful GPR data to be added would require knowledge on where the true ice thickness distribution deviates most from the distribution in the glaciological model, but this would require advanced prior knowledge about the ice thickness that one wants to measure. The problem can be tackled nevertheless by making some specific assumptions (see below).

With our investigations, we address the following questions.

1. Was the experimental geometry and the amount of data acquired in the three investigation areas adequate?

2. Do better experimental layouts exist for constraining the ice thicknesses in a cost-optimized manner?

3. Can some general recommendations be made for designing helicopter-borne GPR surveys on glaciers?

Due to the lack of knowledge on the true ice thicknesses, we assumed that the GlaTE inversion results, shown in Figures 3, 5 and 6 are a good proxy for the actual thickness distributions. Without GPR data, the state of knowledge is represented by the glaciological model (Figures 2a, 5a and 6a). For these models, only 16% (Morteratsch), 10% (Plaine Morte) and 23% (Dom) of the GPR data constraints satisfy the condition $\left\| \mathbf{Gh}^{\mathbf{glac}} - \mathbf{h}^{\mathbf{GPR}} \right\| < \mathbf{\varepsilon}^{\mathbf{GPR}}$, and the average ice thickness misfits over the entire glacier area $\left( mean\left( \mathbf{h}^{\mathbf{glac}} - \mathbf{h}^{\mathbf{true}} \right) \right)$ ($\mathbf{h}^{\mathbf{true}}$ = "true" model) are 20 m, 25 m and 15 m for the three data sets, respectively. It should be noted that the glaciological models are calibrated with $\alpha_{GPR}$. If no GPR data would have been available, the performance of the glaciological models would be even worse.

Subsequently, it is analyzed which of the profiles $j$ $(j = 1…number\ of\ profiles)$ causes the largest discrepancies between $\mathbf{h}^{\mathbf{GPR}}$ and $\mathbf{h}^{\mathbf{glac}}$. For that purpose we define

$$(12) \qquad d_1^{cost} = \max_j \left( \frac{\sum_{i=1}^{i=n_j} P\left( \left| h_{ij}^{GPR} - h_{ij}^{glac} \right| \right)}{\mathbf{c}_j} \right),$$






where index $i$ runs over all $n_j$ data points of profile $j$. $h_{ij}^{GPR}$ and $h_{ij}^{glac}$ represent the
measured and modelled ice thickness at data point $i$ of profile $j$. The function $P$ is defined
as

(13)
$$P(x) := \begin{cases} 1 & if \ x > \varepsilon^{GPR} \\ 0 & if \ x \le \varepsilon^{GPR} \end{cases} .$$


Since longer profiles would be associated with higher (monetary) data acquisition costs,
the discrepancy $d_1^{cost}$ is normalized with a cost factor $c_j$, defined as

(14)
$$c_j = \max\left(len_j, 200\right) ,$$


where $len_j$ represents the length of profile $j$. This cost function assumes that the
acquisition costs increase linearly with profile length, which is realistic, because the
helicopter costs are typically charged per minute of flight time. To avoid that overly short
profiles would dominate $d_1^{cost}$, the assumption was made that profiles with $len < 200$ m
would incur the same costs (for such short profiles the flight time is typically governed
by positioning the helicopter at the starting point of a profile).

The profile associated with the largest discrepancy $d_1^{cost}$ is expected to offer the largest
amount of additional information per unit cost. In this virtual experiment, we assumed
that one would acquire this profile and subsequently perform a GlaTE inversion, yielding
an improved model $\mathbf{h}^{\text{est}_k}$. Index $k$ indicates the actual state of the experimental design,
that is, $k$ is equal to 1, when adding the first profile. Then, the next profile line to be
acquired is identified using

(15)
$$d_{k+1}^{cost} = \max_j \left( \frac{\sum_{i=1}^{i=n_j} P\left(\left|h_{ij}^{GPR} - h_{ij}^{est_k}\right|\right)}{c_j} \right)$$


Repeated application of Equation (15) identifies an optimized sequence for how the
profiles should be acquired. Figures 7a, 7c and 7e show the evolution of what we call the
"data fit curve", i.e. the evolution of

435 (16)
$$d_{k+1}^{fit} = \max_j \left( \frac{\sum_{i=1}^{i=n_j} \hat{P}\left(\left|h_{ij}^{GPR} - h_{ij}^{est_k}\right|\right)}{n_j} \right)$$



with

(17)
$$\hat{P}(x) := \begin{cases} 0 & \text{if } x > \varepsilon^{GPR} \\ 1 & \text{if } x \le \varepsilon^{GPR} \end{cases}.$$

For the Morteratsch and Plaine Morte data, there is an approximately linear increase of
the data fit curve. Likewise, we observe a corresponding linear decrease of the average
model misfit. As discussed in Maurer et al. (2010), benefit-cost curves, such as the $\boldsymbol{d}^{fit}$
graphs in Figure 7, typically enter into the area of diminishing returns at some stage, that
is, the curves exhibit a characteristic kink and flatten out at larger numbers of profiles.
This indicates that it becomes increasingly expensive to obtain additional information.
The curves in Figures 7a and 7c therefore indicate that the area of diminishing returns
was not reached during the Morteratsch and Plain Morte campaigns, and that it would
have been useful to acquire more profiles. In contrast, the $\boldsymbol{d}^{fit}$ and average misfit curves
for the Dom region (Figure 7e) start flattening out, although we do not observe a
characteristic kink in the curves. This indicates that it would have been very expensive to
obtain a more accurate ice thickness distribution for the Dom field site.

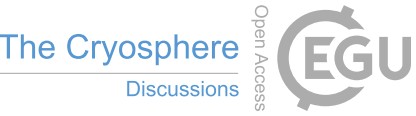



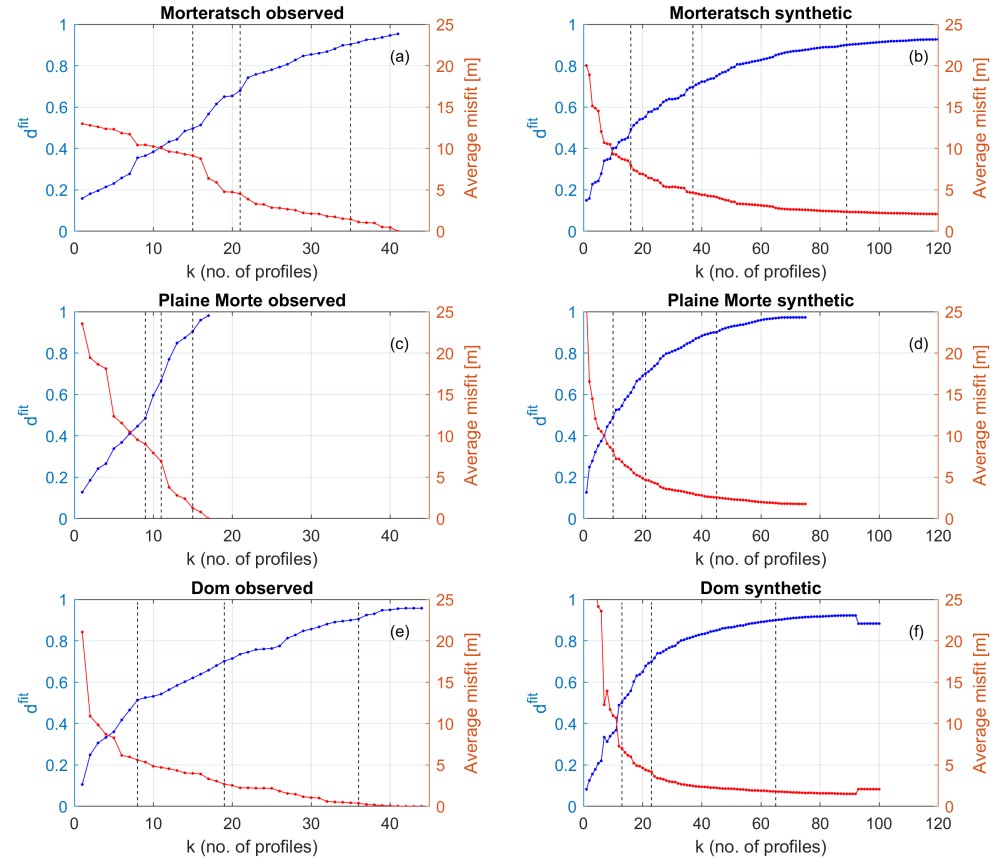

*Figure 7: Evolution of data fit $\mathbf{d}^{\text{fit}}$ (blue curves) and average data misfit*

$mean\left(\mathbf{h}^{\text{est}_k} - \mathbf{h}^{\text{true}}\right)$ *(red curves). Panels a), c) and e) show the results for the observed*

*data, and panels b), d) and f) show the results for the synthetic data generated on a*

*densely spaced grid of hypothetical profiles . Vertical dashes lines indicate the number of*

*profiles required to achieve $d^{fit}$ values of 0.5, 0.7 and 0.9 (see also Figures 8 to 13).*

Figures 8 to 10 show examples of model misfit plots $\left(\mathbf{h}^{\text{est}_k} - \mathbf{h}^{\text{true}}\right)$ superimposed with the
selected profile lines. The corresponding stages of the selection procedure are indicated
with black dashed lines in Figures 7a, 7c and 7e. For the Morteratschgletscher, profiles
are selected preferentially in the western part, because the model fit is already quite good
in the eastern region. For Plaine Morte, it is interesting to note that most N-S profiles are
selected before the longer and thus more expensive E-W oriented profiles are considered.
In the Dom region, no obvious selection patterns can be recognized.





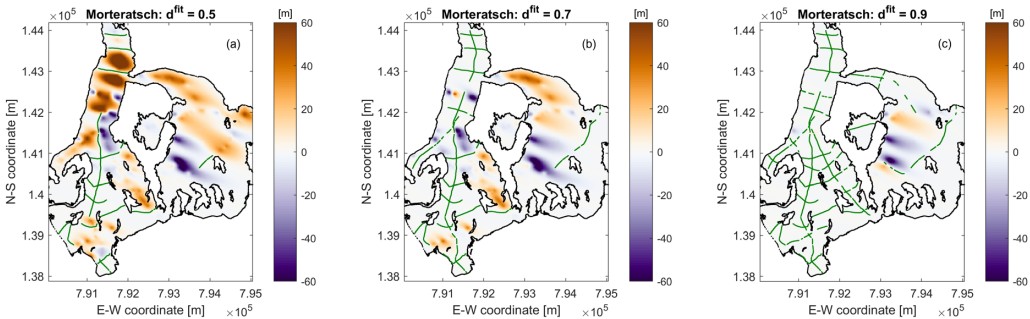


*Figure 8: Morteratsch model misfit* $\mathbf{h}^{true}$ *-* $\mathbf{h}^{est_k}$ *after selected stages of the experimental design procedure using observed data (see also vertical dashed lines in Figure 7). The selected GPR profiles are superimposed with green lines.*

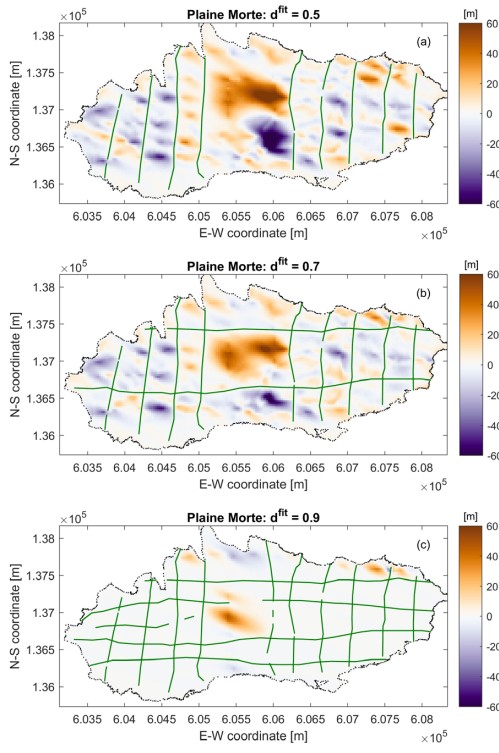


*Figure 9: Plaine Morte model misfit* $\mathbf{h}^{true}$ *-* $\mathbf{h}^{est_k}$ *after selected stages of the experimental design procedure using observed data (see also vertical dashed lines in Figure 7). The selected GPR profiles are superimposed with green lines.*





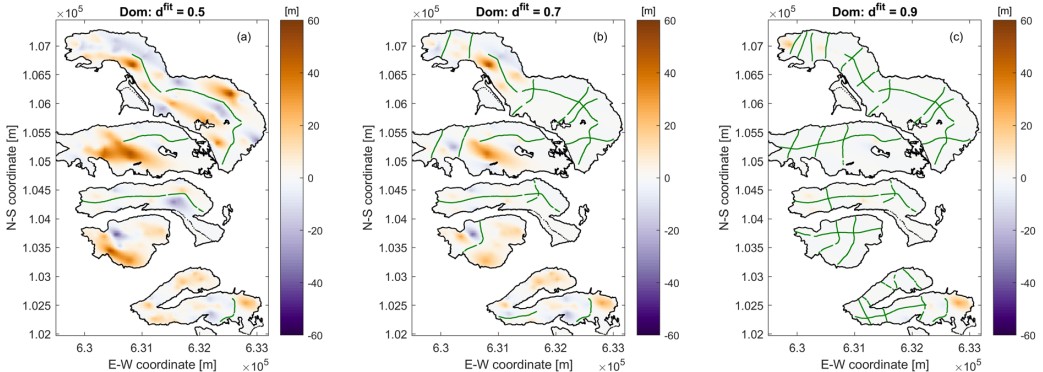

*Figure 10: Dom model misfit* $\mathbf{h}^{true}$ - $\mathbf{h}^{est_k}$ *after selected stages of the experimental design*
*procedure using observed data (see also vertical dashed lines in Figure 7). The selected*
*GPR profiles are superimposed with green lines.*
A major limitation of this design experiment is that the "true" model and the recorded
GPR profiles have a strong dependency. When all profiles of a particular region are
selected, there is a perfect match between $\mathbf{h}^{est_k}$ and $\mathbf{h}^{true}$. However, this is the result of
our choice of the "true" model, and thus not indicate that this data set is optimal. To
reduce, at least partially, this dependency, we have generated synthetic data sets that are
covering all glacierized areas with a dense grid. We assumed a line spacing of 100 m and
an inline sampling interval of 0.5 m, which is representative for the helicopter-borne GPR
data that we acquired. With such a comprehensive data set, the experimental design
procedure should have more flexibility to choose cost-optimized suites of profiles.
The resulting benefit-cost curves are shown in Figures 7b, 7d and 7f. As expected, the
curves start flattening out after selecting a sufficiently large number of profiles. For the
Morteratschgletscher (Figure 7b), it seems to be worthwhile acquiring more than the 43
profiles acquired during the actual experiment. After about 70 profiles, there is no
significant benefit observed. Likewise, the curves for the Glacier Plaine Morte (Figure
7d) indicate clearly that acquiring a larger number of profiles would have been beneficial.
After adding about 40 profiles, the $\mathbf{d}^{fit}$ curve starts flattening out. Only for the Dom
region, the amount of profiles chosen for the actual survey seems to be adequate (Figure
7f). Note that the decrease in $\mathbf{d}^{fit}$ at about $k = 8$ and $k = 90$ in Figure 7f are the result of
the applied smoothing constraints interfering with the data fit, but this does not affect the
general shape of the curve.
Using the $\mathbf{d}^{fit}$ curves in Figure 7 seems to be a good option for selecting an appropriate
number of profiles, but it is also insightful to consider the associated model misfit curves.




Figures 7b, 7d and 7f indicate that the average misfit $\varepsilon^{GPR} = 5m$ is typically reached well
before the $\mathbf{d}^{\text{fit}}$ curves start flattening out.
For the experimental design with the synthetic data, Figures 11 to 13 shows examples of
model misfit plots $\left( \mathbf{h}^{\text{est}_k} - \mathbf{h}^{\text{true}} \right)$ superimposed with the selected profile lines. In contrast
to the selection based on observed data from the Morteratschgletscher (Figure 8), the
design based on the dense synthetic grid (Figure 11) yields a better balance of profiles
among the eastern and western portions of the glacier. This is the consequence of the
larger flexibility of choosing profiles with the dense grid. For the Glacier Plaine Morte
(Figure 12), it is interesting to note that exclusively N-S oriented profiles were chosen. In
contrast, predominantly E-W oriented profiles were chosen for the Dom region (Figure
13). Both observations are governed primarily by the cost factor $c_j$ in Equation (15).

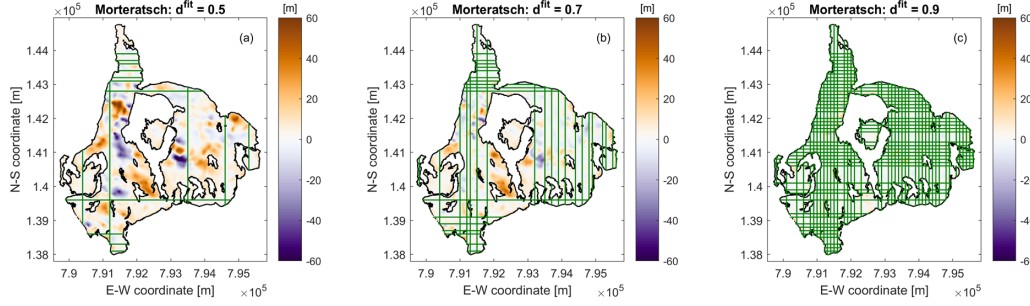

*Figure 11: Morteratschgletscher model misfit $\mathbf{h}^{\text{true}} - \mathbf{h}^{\text{est}_k}$ after selected stages of the*
*experimental design procedure using synthetic data (see also vertical dashed lines in*
*Figure 7). The selected GPR profiles are superimposed with green lines.*





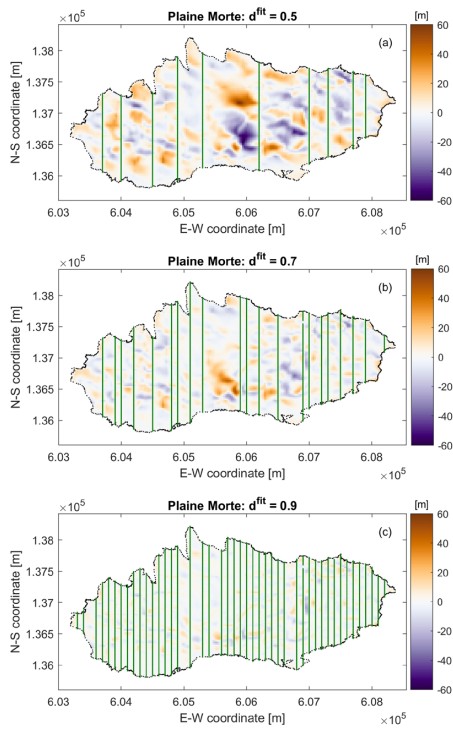


*Figure 12: Glacier Plaine Morte model misfit* $\mathbf{h}^{\text{true}}$ - $\mathbf{h}^{\text{est}_k}$ *after selected stages of the*
*experimental design procedure using synthetic data (see also vertical dashed lines in*
*Figure 7). The selected GPR profiles are superimposed with green lines.*

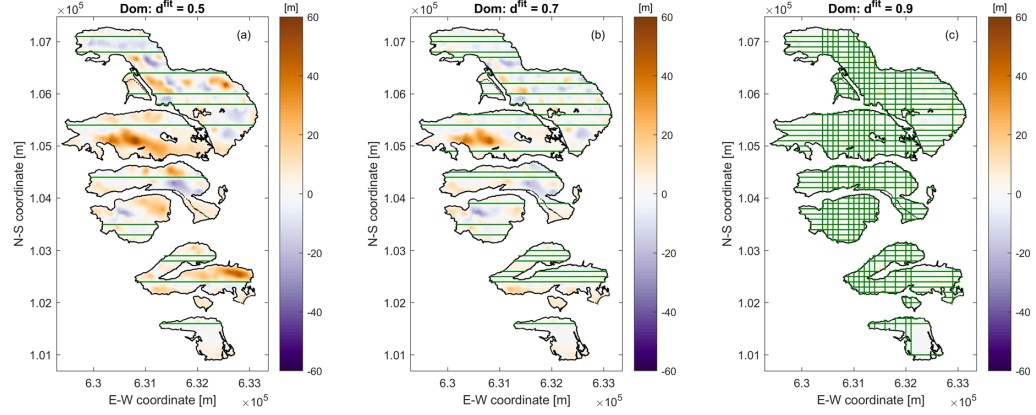


*Figure 13: Dom Region model misfit* $\mathbf{h}^{\text{true}}$ - $\mathbf{h}^{\text{est}_k}$ *after selected stages of the experimental*
*design procedure using synthetic data (see also vertical dashed lines in Figure 7). The*
*selected GPR profiles are superimposed with green lines.*



## 4 Discussion and conclusions
The GlaTE inversion scheme presented in this paper offers numerous beneficial features.
Its main advantage is its versatility, as there are several parameters, by which the
algorithm can be tuned to the peculiarities of a particular investigation area. However,
this is also one of the method's major drawbacks, since the choice of the control
parameters may include a considerable amount of subjectivity. This applies primarily to
the choice of the weighting factors $\lambda_1$, $\lambda_2$ and $\lambda_4$. Finding an appropriate value for $\lambda_4$
can be particularly awkward, since there is typically no ground-truth information
available on the lateral smoothness of the ice thickness distribution. Therefore, we have
chosen to keep $\lambda_1$ and $\lambda_2$ fixed and to determine $\lambda_4$ automatically. Quantifying our
(relative) confidence in the GPR constraints ($\lambda_1$) and glaciological constraints ($\lambda_2$) is
also a non-trivial task. For this problem, however, some physical arguments may exist.
Nevertheless, it might be helpful to repeat the GlaTE inversions with a range of $\lambda_1 / \lambda_2$
ratios and to check the corresponding variations in the resulting models.
Another potential problem is the determination of the scaling factor $\alpha_{GPR}$ in Equation (7).
It is largely dependent on the available GPR data, and it is assumed that the GPR profiles
have a good areal coverage, which might not be always the case. If values for $\alpha_{GPR}$
would be available for a large number of glaciers, a statistical analysis might be used to
correlate the values with specific features of the glaciers (e.g., average steepness,
elevation above sea level, size or shape of the glacier, exposure, etc.). This may be
helpful in areas, where the GPR data coverage is poor or even non-existent.
In principle, any observations (e.g., boreholes) can be employed as data constraints in
Equation (1), but GPR measurements are typically the main source of information.
Migration of the GPR data allows the bedrock reflections to be imaged at the correct
positions and slopes along a profile, but it is possible that the reflections originated from
locations away from the profile lines (off-plane reflections). This may cause systematic
errors affecting the reliability of the results. We note, however, that this is not a problem
specific to GlaTE, but rather a general issue affecting GPR data acquired on a sparse grid.
As mentioned in Section 2, the system of equations in (11) can be augmented by any
linear constraints. An obvious, and in our view particularly useful set of constraints
would be offered by surface displacement measurements. They can be obtained from
differential satellite images and offer full coverage over a glacier. Such constraints could
possibly substitute the smoothness constraints in Equation (11) with a physically more
meaningful quantity.
Despite the limitations of our approach, we judge that our results provided useful insights
for designing GPR experiments, and some answers to the questions posed in Section 3
can be provided.






1.  *Was the experimental geometry and the amount of data acquired in the three*
*investigation areas adequate?*
The benefit-cost curves in Figure 7 indicate that, at least for the Morteratsch and
Glacier Plaine Morte, it would have been useful to acquire more data.

2.  *Do better experimental layouts exist for constraining the ice thicknesses in a cost-*
*optimized manner?*
The experimental layouts in Figures 8 to 13 do not provide unexpected features,
but indicate that acquiring a larger number of shorter profiles, instead of recording
a few long ones, could be beneficial, but it should be noted that we do not take
into account the flight time required to move to the next profiles. This could be
significant on glaciers with steep mountain flanks.

3.  *Can some general recommendations for designing helicopter-borne GPR surveys*
*on glaciers be made?*
Based on our results, it is difficult to offer general recommendations. For
estimating the overall amount of data to be collected, the benefit-cost curves are
most indicative. However, in our case studies they do not flatten out clearly,
thereby indicating that it would be worthwhile acquiring more data. When high-
precision ice thickness maps are required, it is therefore advisable to acquire as
much data as can be afforded.
It is common practice to acquire crossing profiles, but from the experimental
layouts, shown in Figure 12, it could be concluded that it is not necessary to
acquire a large amount of crossing profiles. From a practical point of view, this
recommendation cannot be fully supported. When the signal-to-noise ratio of the
GPR profiles is poor, it can be difficult to identify the bedrock reflections
unambiguously.
It is not realistic to adopt a real-time experimental design strategy (i.e., choosing
the next profile based on the results of the previously acquired data), as assumed
in our virtual experiments in Section 3. However, if logistically feasible, it might
be useful to employ a two-step acquisition strategy. Initially, only a few profiles
could be acquired. After analyzing these data sets, regions, where large
discrepancies between $\mathbf{h}^{est}$ and $\mathbf{h}^{glac}$ exist, could be identified, and a suitable set
of additional profiles could be acquired with a second campaign.



## Acknowledgments

We thank Patrick Lathion, Philipp Schaer and Kevin Délèze from GEOSAT SA, Patrick
Fauchère from Air Glacier, Hansueli Bärfuss from Heli-Bernina, as well as Lasse
Rabenstein and Lino Schmid for acquiring the data. Furthermore, we thank Matthias
Huss for fruitful discussions and Daniel Farinotti for an insightful in-house review, which
improved the clarity of the manuscript. Financial support was provided by ETH Zurich
(Grant ETH-15 13-2), the Innosuisse program SCCER-SOE (Swiss competence center
for energy research, supply of electricity), and the Swiss Geophysical Commission and
ETH Zurich.

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
