# Peer review of "Glacier thickness estimations of alpine glaciers using data and modeling constraints"

_The Cryosphere, 2019_

## Referee Comment (RC1) · Ben Pelto (Referee) · 17 Apr 2019

Langhammer and colleagues present a new algorithm to optimally combine modeled ice thickness and measured ice thickness using an inversion procedure to obtain distributed ice thickness. They also introduce a method to cost-optimize helicopter-borne GPR survey design. I found this manuscript well thought out, well detailed, and relevant. There are a few points that require greater explanation or justification critical to the results of this study. Both deliverables of this project, the GlaTE algorithm for combining model results and GPR measurements, and the sequential optimized experimental design for determining cost-optimized GPR survey transects contribute scientifically to

methods pertaining to glacier ice thickness. The manuscript does not mention whether the GlaTE algorithm will be made publicly available, yet both sections of this manuscript primarily offer value as tools for planning future studies of ice thickness and optimizing data collected in such studies. I would recommend the paper to be considered for publication after consideration of the following points and a list of suggestions for minor corrections and clarifications:

Will GlaTE be made publicly available? This manuscript is presented as a methods paper introducing two frameworks, one (GlaTE algorithm) for creating distributed ice thickness from GPR measurements and model results, and one (SOED techniques) for cost-optimizing GPR surveys. For this study to be of greatest value it is critical these methods are made available so that others may use them.

$E\_GPR$ was estimated to be 5 m. I can find no justification for this value. Five meters may be a reasonable value for $E\_GPR$, but this value assumes great importance in the system of equations L203-228, and is used to draw conclusions: e.g. L316, L392-394. While the point of lines L392-394 would likely remain unchanged, the percentages listed in L392-394 would undoubtably be higher were $E\_GPR$ higher. A low value (5 m) may tend to overfit the available GPR data despite the attempt to avoid doing so via the system of equations. Literature examples of estimated uncertainty in GPR measurements (one method to determine $E\_GPR$):

From Gartner-Roer et al 2014: "A direct comparison of radar measurements with hot water drillings and borehole electrodes indicated that thicknesses derived from radar measurements are usually within $\pm$ 5% of the measured ice thickness (Haeberli and Fisch, 1984). Fischer (2009) estimated the uncertainty of thicknesses for the Austrian glaciers as 5–10% of the measured value. Zamora et al. (2009) applied airborne GPR on Tyndall glacier (Patagonia, Argentina) and observed a maximum ice thickness of 670 m, with an accuracy of $\pm$ 50 m when comparing the results to existing thickness information."

From Sanders et al. 2010: "Comparison with the depths of six boreholes drilled through the glacier (using a hot-water system) showed that the thickness map obtained by radar was generally accurate to better than 10 percent of the ice depth (7 +/- 4 m). Near the steeply-sloping marginal walls, however, where the ice is thin and the topography complex, errors increased to 13 +/- 2 m (27 +/- 8%). The survey revealed an average glacier depth of about 70 m, and maximum depth of about 185 m."

From Marcer et al. 2017: "The error of the interpretation of GPR profiles was evaluated through quantification of the residuals in ice thickness at all intersecting profiles, that is, by crossover analysis. The total number of intersections is 16, and the standard deviation of the differences is 1.9 m."

Specific comments:

L39 Change recourses to resources.

L52 RES acronym not necessary as there is currently no usage throughout the document.

L58 'Un-aliased' was previously written as 'unaliased'.

L95 Awkward phrasing.

L105 Is this regular grid (R) a 10 m grid with the same posting as the resampled DEM described in L264?

L122-131 Were the mass balance/height change estimates (\dot{b} and dh/dt) computed via eqn. 3 and the method of Farinotti et al. (2009) in broad agreement with regional height/mass change gradients as measured by other studies? This region contains a wealth of geodetic and glaciological mass balance observations with which to verify.

L244 Please list glacier area for this site, and at least for the Hohbärggletscher (L250). Perhaps it would be useful to have a small table with glacier attributes, name, size,

range etc.

L245 Should this: '…with slope angles less than 4ĔŽ' be rephrased to average slope angle? It is likely that some of the surface slope of the glacier exceeds 4ĔŽ, particularly on the 'short glacier tongue'.

L254-257 It is stated that the GPR data are a composite of several campaigns. For a given site were these campaigns all within the same year? If over multiple years for a given site, was height change due to mass change accounted for in any way? This may be a small impact on ice thickness, but not given a 5 m ÆŘGPR threshold.

L328, 353,358 Perhaps it would be valuable to have a table containing ice thickness information for each glacier e.g. maximum + mean measured thickness, number of point obs., km of transects, etc.

L454 Excellent figure.

L603-608 These lines accurately depict that cross profiles are of value. Some studies use them to assess GPR profile interpretation error (e.g. Marcer et al. 2017). Would it not be of value to be able to designate a minimum amount of cross profiles in the cost-optimization scheme? As the authors point out, in areas with poor signal to noise ratio, having cross profiles are very valuable. As cross profiles are always built into GPR survey designs, a cost-optimization which may not include them is of lesser value. One could simply add a few cross profiles to the SOED output, but if the SOED were allowed to incorporate these transects, perhaps the SOED technique would change the designed survey.

L629 Will these GPR data be added to the Glacier Thickness database, GlaThiDa (Gärtner-Roer et al., 2014)?

Gärtner-Roer, I., Naegeli, K., Huss, M., Knecht, T., Machguth, H. and Zemp, M.: A database of worldwide glacier thickness observations, Glob. Planet. Change, 122, 330–344, doi:10.1016/j.gloplacha.2014.09.003, 2014.

Marcer, M., Stentoft, P. A., Bjerre, E., Cimoli, E., Bjørk, A., Stenseng, L. and Machguth, H.: Three decades of volume change of a small Greenlandic glacier using ground penetrating radar, Structure from Motion, and aerial photogrammetry, Arct. Antarct. Alp. Res., 49(3), 411–425, 2017.

Sanders, J., Cuffey, K., MacGregor, K., Kavanaugh, J. and Dow, C.: Dynamics of an alpine cirque glacier, Am. J. Sci., 310(8), 753–773, 2010.

Zamora, R., Ulloa, D., Garcia, G., Mella, R., Uribe, J., Wendt, J., Rivera, A., Gacitúa, G. and Casassa, G.: Airborne radar sounder for temperate ice: initial results from Patagonia, J. Glaciol., 55(191), 507–512, doi:10.3189/002214309788816641, 2009.

---

## Referee Comment (RC2) · Douglas Brinkerhoff (Referee) · 2 May 2019

**Synopsis**

In "Glacier thickness estimations of alpine glaciers using data and modeling contraints", Langhammer and colleagues present a new mathematical formulation of a long-standing problem. They formulate the problem of physics-based interpolation to finding ice thickness values between radar flight lines as a system of linear equations, and perform an exploration of the hyper-parameters that can be adjusted to yield different results. This system of equations includes components representing the contribution of observations, physical constraints based on the shallow-ice approximation, geometric constraints on glacier extent, and regularization of spatial gradients. The authors apply their method to the problem of determining an ideal distribution of expensive ice thickness observations, yielding guidance on how to construct GPR campaigns.

**Comments**

Thickness Estimation Method

While the explicit formulation of the problem as a sparse system of equations is new, each component of the model is not. Looking as far back as Morlighem, 2011, the problem is specified as a minimization problem in which there exists a data misfit function over flightlines, a physical misfit function over the entire glacier domain, and a spatial regularization to impose smoothness. The difference here is the substitution of the shallow ice approximation for mass-conservation, which alleviates the need for velocity and surface mass balance observations. We note that this physical model was already developed for use in physics-based interpolation by Farinotti and Huss (2009), including an application in which corrections based on GPR were performed (Huss and Farinotti, 2014). There is nothing inherently problematic in replication of previous methods: however, it would be useful to have a specific discussion of how GlaTE is different from methods to solve this problem that have gone before.

The exploration of the weights on specific model components lacks sufficient rigor. Why are the various values of $\lambda$ set the way that they are? It makes little sense to explore these parameters heuristically, since they have a clear probabilistic meaning ($\lambda = \frac{1}{\sigma}$, which is to say that the equations should be weighted in inverse proportion to measurement/model uncertainty). Such a probabilistic formulation of the problem was explored in Brinkerhoff et al. (2015). I would like to see more of an effort to place the

various $\lambda$ in a real-world context so that they can be specified objectively.

Despite the criticism of the last paragraph, I think that the explicit nature by which the strengths of the various model objectives are imposed produces a consistent and efficient platform for exploring modelling choices. The linear nature of the system of equations means that a large number of realizations of ice thickness could be generated for different parameter choices, which the authors acknowledge in Section 4 is both an advantage and a drawback. Given the model's efficiency, why not take the next step in determining hyper-parameter values and run cross-validation on held-back radar flightlines? This is almost what is done in the Experimental Design section, but not quite. This procedure would capitalize on the model's strengths, and would also yield a sort of guidebook on how the algorithm might be used without having to make a lot of choices about $\lambda$ that might be only marginally defensible.

Finally, I'm confused about the computation of $\alpha$. This method seeks to find alpha that yields a mean misfit which is as close to zero as possible. However, this admits very large pointwise deviations between modelled and observed thickness. Perhaps a better metric might be sum squared error between modeled and measured thickness. Better yet, instead of optimizing on $\alpha$, why not minimize with respect to , $A$, and/or $n$ directly? Uncertainty in these values is the reason behind introducing $\alpha$, yet their combined influence is only poorly captured by a linear approximation.

Experimental Design Procedure

The methods presented in this section are both novel and make good sense, and yield interesting insights into the degree of coverage necessary to yield a good model of the glacier bed. The idea of sequentially finding the profile that would yield the greatest change from an unconstrained inversion method is very general and could be applied to all types of physical models.

One thing that would help to understand what lessons might be learned from this analysis is a more in depth discussion of the nature of the automatically selected profiles. Why, for example, in the synthetic case is there a dominant flightline orientation that differs between glaciers? I would suspect it has something to do with the relative information content of cross- versus along-slope profiles, but it's hard to say. The authors are in a good position to explore this question more fully, and answering the question of which orientation is better for constraining glacier thickness would be an important advance.

---

## Referee Comment (RC3) · Fabien Maussion (Referee) · 14 May 2019

In this manuscript, Langhammer and colleagues present a new method to compute the distributed ice thickness of glaciers under observational constraints. I enjoyed reading the manuscript, and especially the second part introduces new methods and insights that could be useful elsewhere. I have a couple of recommendations below (I wrote them without reading the two other reviews: you might notice some overlap).

[Figure]

**General comments**

**Calibration of the regularization parameters and model validation**

This is the part that most confuses me. I'm not sure as to what the respective $\lambda_i$ experiments tell us, and I wonder if there could be a more systematic, quantitative way to calibrate these parameters. The first method that comes to mind is to use cross-validation instead of the current "step-test-stop" implementation. As an example from the machine learning literature, implementations of regularization in LASSO algorithms often use cross-validation to determine $\lambda$. In your case, you have three free parameters which are likely to compensate each other, but currently we actually don't know if this is the case or not.

This brings me to the second point: at line 300, you write: *"we conclude that the GlaTE inversion approach works well"*. But how do we know this, and what does "work well" mean? Since the lambda parameters are chosen in a way that almost all GPR measurements are fit perfectly, yes the model "works well" but it might be over-fitted despite of the other constraints (for example, in one experiment $\lambda_2$ even goes to zero, meaning that the algorithm becomes a mere interpolation). Here again, cross-validation could help to assess the robustness of the model for regions where no GPR data is available (note that it should be another cross-validation loop than the one used to calibrate $\lambda$, i.e. with truly unseen data).

Given my late review and the fact that I don't know how computationally demanding your inversion method is, the comments above can be understood as a "recommendation" more than a "must do". I believe however that the question: "how well does GlaTe really work with unseen data" should be addressed at L300 or in the discussion.

**Code and data availability**

According to this journal's data policies (https://www.the-cryosphere.net/about/data_policy.html), code and data should be made available if possible. I cannot enforce
these rules but I strongly recommend them: are your GPR data available, and if yes where? The GlaThiDa database would be a good place for them, although they do no guarantee attribution. Would you consider sharing your model code under an open-source license? I believe that both would be a strong asset for the community and would increase the visibility of your work.

**Specific comments**

**L117** Glacier flowsheds. Do the flowsheds create discontinuities in the apparent MB field and therefore in h? I would assume they do, e.g. at a junction between two glacier branches. Is this a problem?

**L131, Eq. 3** to compute the apparent MB with the equilibrium assumption you still need to use a specific function for the MB. E.g. linear, or linear with with two slopes, etc. Give more details about what is tuned here. In particular, mention what you do in the case of the glacier cluster case study: I assume that $\Omega_G$ is computed for each entity independently? And what about $\alpha_G$, is it the same for all glaciers?

**L147** "lower boundary of Di". How is this computed? Is this equivalent to the grid spacing of the gridpoint i?

**Equation 6** which slope is $\theta$? Is it the same as $\phi$, introduced above?

**L167** parameter $\alpha$. Another approach would be to use $\alpha$ as a correction factor for the uncertain parameters, for example A. In this case the calibration function is not linear anymore, but it would be more physically consistent with the uncertainties in A.

**L170** shouldn't the equation be mean(abs(diff)), i.e. the mean absolute deviation (MAD)? With your formulation you are only minimizing the overall bias, which

would allow strong deviations at individual points. (but maybe this is what you intended).

**Solver** can you add an example about the dimensionality of the LSQR problem in one of the three cases, and of the time needed to solve it?

**Figure 1** Text and legend seem to have inverted b) and c)

**L431** if I understand well, the distance between profiles is not taken into account, right? So the flight time from one profile to another does not enter the cost function?

**L553-L557** just a comment: as someone who actually tried to do this, let me say that it is very unlikely to work like that... I'd be glad to be proven wrong though!

---

## Author Comment (AC1) · 14 Jun 2019

see PDF enclosed

Please also note the supplement to this comment:
https://www.the-cryosphere-discuss.net/tc-2019-55/tc-2019-55-AC1-supplement.pdf

---

## Author Comment (AC2) · 14 Jun 2019

Dear Editor and reviewers,

Thank you for all the constructive comments on our manuscript. We feel sure that addressing the comments will improve the quality of the paper. Below, we provide our responses to all three reviewer comments.

Best Wishes

Hansruedi Maurer (on behalf of the author team)

**Reviewer 1 (Ben Pelto)**

*General comments*

- *Availability of code and data*
  We plan to make the GlaTE Matlab scripts publically available on GitHub. Likewise, we will upload the data sets employed in the paper on this platform. This will allow reproducing all our results, and we hope that the codes will be helpful for other data sets.
- *Accuracy of H-GPR ice thickness estimates*
  Indeed, the accuracy of the H-GPR thickness estimates is critical for our algorithm. As noted correctly by the reviewer, the literature offers quite a range of thickness estimates. We have re-evaluated our data and concluded that a depth-dependent accuracy (i.e., percentage error) would be a better option. A reasonable choice for our data sets is 5%, that is, an accuracy of 5 m would correspond to a thickness of 100 m. When available, it would be straightforward to consider individual accuracy estimates for the individual data points. In the modified manuscript we include a more detailed discussion on this topic.

*Specific comments*

- *Editiorial comments*
  We have addressed all editorial comments
- *Sampling of DTM*
  Yes, the DTM is sampled on R, as it was indicated on line 105
- *Mass balance estimates*
  Yes, the results are in broad agreement with typical values obtained in this region
- *Table with glaciers*
  The revised paper includes a table with the important characteristics of the glaciers considered
- *Merging of different campaigns*
  An earlier version of the manuscript included data sets of merged campaigns. However, we decided to show only data sets that were acquired in the framework of a single campaign, to avoid the problem of the ongoing melt. The statement about the merged data set was just a remnant from the earlier draft, and we have removed it in the revised version.
- *Table with ice thickness estimates*
  Since we make the data sets publically available, we don't think that such a table is necessary
- *SOED and crossing profiles*
  We added additional text in the revised manuscript to address this issue

- ***Adding data to GlaThiDa***
  Our measurements in Switzerland until 2015 are covered in the GlaThiDa 3.0 release, and we intend to provide an update with the next release

**Reviewer 2 (Douglas Brinkerhoff)**

*General comments*

- ***Novelty of approach***
  We do not claim that the ice thickness estimation approach within GlaTE is novel. As indicated on line 112ff, any of the algorithms described in the literature can be incorporated. The novelty lies rather in the consideration of the uncertainties of the H-GPR measurements, and in the formulation in form of a sparse system of linear equations, which allows incorporating any further constraints. In the revised manuscript, we make this more obvious in the abstract.
- ***Choice of weighting parameters $\lambda_1$ to $\lambda_4$***
  We agree with Reviewers 2 and 3 that the discussion on the choice of the weighting parameters may be confusing. Based on their comments, we re-thought the strategy for choosing $\lambda_1$ to $\lambda_4$. The revised manuscript includes a more detailed description. In brief, we fix $\lambda_3$ to a constant value. This parameter has very little effect on the inversion result. Next, we perform a series of inversions with different $\lambda_1/\lambda_2$ ratios (remain fixed during a single inversion run). During each inversion run, the smoothing parameter $\lambda_4$ is gradually lowered, until a prescribed percentage (e.g., 95%) of the GPR data is fitted with the prescribed accuracy. When the $\lambda_1/\lambda_2$ ratio is getting too small, the inversion algorithm fails to match the GPR data, even when $\lambda_4 = 0$. The lowest ratio, which allows to fit the GPR data, is finally chosen. This procedure (i) allows to fit the GPR data with a prescribed accuracy (no overfitting), (ii) maximizes the contribution of the glaciological constraints and (iii) minimizes the influence of the (unphysical) smoothing constraints.

  Reviewers 2 and 3 suggested cross validation methods for identifying optimal weighting parameters. This is potentially an interesting option, but we judge the procedure outlined above to be physically more meaningful and computationally cheaper.
- ***Choice of $\alpha$ parameter***
  Reviewer 2 is right. It makes conceptually much more sense to minimize the squared differences between observed and modelled thicknesses. We changed the manuscript accordingly and recomputed the three test cases. Interestingly, the $\alpha$ values, obtained with the new procedure, are very similar to the old values.
- ***Choice of lines during SOED procedure***
  The choice of the lines is influenced by a plethora of factors. However, the procedure does not consider (explicitly) the amount of crossing profiles, which could be advantageous for cross-checks. We have added a more detailed explanation on this topic.

**Reviewer 3 (Fabien Maussion)**

*General comments*

- **Choice of regularization parameters**
  See response to Reviewer 2 on this topic.
- **Objective assessment of GlaTE performance**
  We agree that our statement concerning the performance of GlaTE is somewhat weak. We have participated in the ITMIX2 initiative, where numerous approaches were compared in form of blind tests. Evaluation of ITMIX2 is still in progress, but we make a reference to this initiative, which is certainly a good measure for the performance of GlaTE.
- **Code and data availability**
  See response to Reviewer 1 on this topic

*Specific comments*

- **Flowsheds**
  We also expected discontinuities between flowsheds, but surprisingly this was not the case.
- **Apparent mass balance computation and glacier cluster**
  More explanation were added to the text
- **Lower boundary of Di**
  We followed the approach of Clarke et al. (2013)
- **$\theta$ vs $\phi$**
  This is the same quantity. The typo was corrected
- **$\alpha$ parameter**
  $\alpha$ accounts for the uncertainties of all multiplicative factors in Equation (5), also including $A$.
- **mean(abs(diff())) issue**
  See corresponding response to Reviewer 2
- **LSQR**
  The system of equations includes ~300,000 rows and ~90,000 columns. Due to the sparseness of the system matrix, the LSQR algorithm requires only about 2 seconds on a standard PC with a 3 GHz processor. However, due to the adjustments of the smoothing parameter $\lambda_4$, the system of equations needs to be solved several times during an inversion run.
- **Figure 1**
  The figure caption (resp. the figure itself) was corrected
- **Flight time to next profile**
  Yes, this is correct. We did not account for the transition time. This was already mentioned on line 590.
- **Statistical analysis for determining $\alpha$**
  During the next few months, we will analyze a very large data set acquired over all significant glaciers in Switzerland. We hope that we can prove you to be wrong …..

---

## Author Comment (AC3) · 14 Jun 2019

see PDF enclosed

Please also note the supplement to this comment:
https://www.the-cryosphere-discuss.net/tc-2019-55/tc-2019-55-AC3-supplement.pdf